# Animal, Fungi, and Plant Genome Sequences Harbor Different Non-Canonical Splice Sites

**DOI:** 10.3390/cells9020458

**Published:** 2020-02-18

**Authors:** Katharina Frey, Boas Pucker

**Affiliations:** 1Genetics and Genomics of Plants, Center for Biotechnology (CeBiTec), Bielefeld University, 33615 Bielefeld, Germany; bpucker@cebitec.uni-bielefeld.de; 2Graduate School DILS, Bielefeld Institute for Bioinformatics Infrastructure (BIBI), Bielefeld University, 33615 Bielefeld, Germany; 3Molecular Genetics and Physiology of Plants, Faculty of Biology and Biotechnology, Ruhr-University Bochum, Universitätsstraße 150, 44801 Bochum, Germany

**Keywords:** splicing, spliceosome, RNA-Seq, gene structure, mRNA processing, introns, sequence conservation, splice site analysis pipeline

## Abstract

Most protein-encoding genes in eukaryotes contain introns, which are interwoven with exons. Introns need to be removed from initial transcripts in order to generate the final messenger RNA (mRNA), which can be translated into an amino acid sequence. Precise excision of introns by the spliceosome requires conserved dinucleotides, which mark the splice sites. However, there are variations of the highly conserved combination of GT at the 5′ end and AG at the 3′ end of an intron in the genome. GC-AG and AT-AC are two major non-canonical splice site combinations, which have been known for years. Recently, various minor non-canonical splice site combinations were detected with numerous dinucleotide permutations. Here, we expand systematic investigations of non-canonical splice site combinations in plants across eukaryotes by analyzing fungal and animal genome sequences. Comparisons of splice site combinations between these three kingdoms revealed several differences, such as an apparently increased CT-AC frequency in fungal genome sequences. Canonical GT-AG splice site combinations in antisense transcripts are a likely explanation for this observation, thus indicating annotation errors. In addition, high numbers of GA-AG splice site combinations were observed in *Eurytemora*
*affinis* and *Oikopleura*
*dioica*. A variant in one U1 small nuclear RNA (snRNA) isoform might allow the recognition of GA as a 5′ splice site. In depth investigation of splice site usage based on RNA-Seq read mappings indicates a generally higher flexibility of the 3′ splice site compared to the 5′ splice site across animals, fungi, and plants.

## 1. Introduction

Splicing, the removal of introns from initial transcripts is an essential step during the generation of mature messenger RNAs (mRNAs) in eukaryotes. This process allows variation, which provides the basis for quick adaptation to changing conditions [1,2]. Alternative splicing, e.g., skipping exons, usage of alternative 5′ or 3′ splice sites, and the retention of introns, results in an enormous diversity of synthesized proteins and, therefore, substantially expands the diversity of products encoded in eukaryotic genomes [3,4,5,6].

The full range of functions, as well as the evolutionary relevance of introns, are still under discussion [7]. However, introns are energetically expensive for the cell to maintain as the transcription of introns costs time and energy and the removal of introns has to be exactly regulated [8]. Dinucleotides at both intron/exon borders mark the splice sites and are therefore highly conserved [9]. GT at the 5′ end and AG at the 3′ end of an intron form the canonical splice site combination on DNA level. More complexity arises through non-canonical splice site combinations, which deviate from the highly conserved canonical combination. Besides the major non-canonical splice site combinations, GC-AG and AT-AC, several minor non-canonical splice site combinations have been detected before [9,10].

Furthermore, the position of introns in homologous genes across organisms, which diverged 500–1500 million years ago, are not conserved [11]. In addition, many intron sequences mutate at a higher rate due to having much less of an impact on the reproductive fitness of an organism, compared to a mutation located within an exon [12]. These factors, along with the existence of several non-canonical splice sites, make the complete prediction of introns, even in non-complex organisms, such as yeast, almost impossible [13,14]. Moreover, most introns that can be predicted computationally still lack experimental support [15].

Splice sites are recognized during the splicing process by a complex of small nuclear RNAs (snRNAs) and proteins: the spliceosome [16]. U2-spliceosome and U12-spliceosome are two subtypes of this complex, which comprise slightly different proteins with equivalent functions [17,18,19]. Although the terminal dinucleotides are important for the splicing process, these splice sites are not sufficient to determine which spliceosome is processing the enclosed intron [20]. Most canonical GT-AG and major non-canonical GC-AG introns are spliced by the major U2 spliceosome [21,22]. While most AT-AC splice site combinations are recognized by the minor U12 spliceosome; some so-called AT-AC type II introns are spliced by the major U2 spliceosome [23,24]. This demonstrates the complexity of the splicing process, which involves additional signals present in the DNA. Even though multiple mechanisms could explain the splicing process, the exact mechanism of non-canonical splicing is still not completely resolved [5].

Branching reaction and exon ligation are the two major steps of splicing [25,26]. In the branching reaction, the 2′-hydroxyl group of the branch point adenosine initiates an attack on the 5′-phosphate of the 5′ splice site [27,28]. This process leads to the formation of a lariat structure. Next, the exons are ligated and the intron is released through attack of the 3′-hydroxyl group of the 5′ exon at the 3′ splice site [25].

Previous in depth analyses of non-canonical splice sites in fungi and animals were often focused on a single or a small number of species [9,29,30]. Several studies focused on canonical GT-AG splice sites but neglected non-canonical splice sites [31,32]. Our understanding of splice site combinations is more developed in plants compared to other kingdoms [10,33,34,35,36,37]. Previous works reported 98% GT-AG splice site combinations in fungi [29], 98.7% in plants [10], and 98.71% in animals [9]. Consequently, the proportion of non-canonical splice sites, other than the canonical splice site GT-AG, is around or below 2% [9,10,29]. To the best of our knowledge, it is not known if the value reported for mammals is representative for all animals. Non-canonical splice site combinations can be divided into major non-canonical GC-AG and AT-AC combinations and the minor non-canonical splice sites, which are all other dinucleotide combinations at the terminal intron positions. The combined frequency of all minor non-canonical splice site combinations is low, e.g., 0.09% in plants, but still exceeds the frequency of the major non-canonical AT-AC splice sites [10]. Despite this apparently low frequency, non-canonical splice site combinations have a substantial impact on gene products, especially on exon-rich genes [10]. Of all plant genes with exactly 40 exons, 40% have at least one non-canonical splice site combination [10].

Consideration of non-canonical splice sites is important for gene prediction approaches, because ab initio identification of these splice sites is computationally extremely expensive and therefore rarely applied [33]. Moreover, as many human pathogenic mutations occur at the 5′ splice site [38], it is of substantial interest to understand the occurrence and usage of non-canonical splice sites. Therefore, several non-canonical splice sites containing AG at the 3′ site were investigated in human fibroblasts [38]. Alongside this, fungi are interesting due to their pathogenic properties and importance in the food industry [39]. Since splicing leads to high protein diversity [3,4,5,6], the analysis of splicing in fungi is important, with respect to biotechnological applications, e.g., development of new products.

Non-canonical splice sites are frequently considered as artifacts [22] and therefore excluded from analyses [31,32]. Further, RNA editing of GT-AA to GT-AG splice sites on RNA level is possible [40]. This leads to the transformation of non-canonical splice site combinations into canonical combinations. Previous studies supported minor non-canonical splice site combinations in single or few species [9,29,30] and systematically across plants [10,33,34,35,36,37]. In this study, a collection of annotated genome sequences from 130 fungi and 489 animal species was screened for canonical and non-canonical splice site combinations in representative transcripts. RNA-Seq datasets were harnessed to identify biologically relevant and actually used splice sites based on the available annotation. Non-canonical splice site combinations, which appeared at substantially higher frequency in a certain kingdom or species, were analyzed in detail. As knowledge about splice sites in plants was available from previous investigations [10,33], a comparison between splice sites in fungi, animals, and plants was performed.

## 2. Materials and Methods

### 2.1. Analysis and Validation of Splice Site Combinations

A detailed technical description of all included scripts with usage examples is available from the corresponding GitHub repository [10]. A wrapper script is included to automatically perform the analyses based on provided genome sequence (FASTA), annotation (GFF3), and RNA-Seq read mapping (BAM) per species of interest.

Genome sequences (FASTA) and corresponding annotations (GFF3) of 130 fungal species and 489 animal species were retrieved from the National Center for Biotechnology Information (NCBI). Representative transcript and peptide sequences were extracted as described before [10]. General statistics were calculated using a Python script [10]. The completeness of all datasets was assessed with Benchmarking Universal Single Copy Orthologs (BUSCO) v3 [41] running in protein mode on the representative peptide sequences using the reference datasets ‘fungi odb9′ and ‘metazoa odb9′, respectively [42] (Appendix A).

To validate the detected splice site combinations, paired-end RNA-Seq datasets were retrieved from the Sequence Read Archive (SRA) [43] (Appendix A). The following validation approach [10] utilized STAR v2.5.1b [44] for the read mapping and Python scripts for downstream processing [45]. RNA-Seq reads were considered mapped if the alignment showed at least 95% similarity and covered 90% of the read length. Splice sites were considered valid if the genomic positions were spanned by at least three reads and showed a coverage drop of 20% when moving from an exon into an intron sequence. Summaries of the RNA-Seq read coverage depth at splice sites in animals [46] and fungi [47] were made available as part of this study.

RNA-Seq read mappings with STAR v2.5.1b and HiSat2 v.2.1.0 were compared based on a gold standard generated by exonerate, because a previous report [48] indicated a superiority of STAR. STAR parameters were set as described above and HiSat2 was applied with default parameters. All transcripts with non-canonical splice sites in *Arabidopsis thaliana* and *Oryza sativa* were considered. When investigating the alignment of RNA-Seq reads over non-canonical splice sites, we observed a high accuracy for both sequence read mappers without a clear difference between them. Previously described scripts [10] were adjusted for this analysis and updated versions are available on GitHub [45] The distribution of genome sizes was analyzed using the Python package dabest [49]. Sequence logos for the analyzed splice sites were designed at http://weblogo.berkeley.edu/logo.cgi [50].

### 2.2. Calculation of the Splice Site Diversity

A custom Python script (splice_site_divergence_check.py) was applied to calculate the Shannon diversity index (H′) [51] of all splice site combinations in fungi, animals, and plants [45]. To determine the significance of the obtained results, a Kruskal–Wallis test [52] was performed using the Python package scipy [53]. Further, the interquartile range of all distributions was examined.

### 2.3. Investigation of a Common Non-Canonical Splice Site in Fungal Genome Sequences

A Mann–Whitney U test implemented in the Python package scipy was performed to analyze differences in the number of minor non-canonical splice site combinations. The observed distributions were visualized in a boxplot [45] constructed with the Python package plotly [54] (ss_combination_frequency_boxplot.py).

### 2.4. Detection of Spliceosomal Components

A potential U1 snRNA of *Pan troglodytes* (obtained from the NCBI; GeneID:112207549) was subjected to BLASTn v.2.8.1 [55] against the genome sequences of selected species. Hits with a score above 100, with at least 80% similarity and with the conserved sequence at the 5′ end of the snRNA [56], were investigated, as these sequences are potential U1 snRNAs. The obtained sequences were compared, and small nucleotide variants were detected.

To assess possible duplications of spliceosomal components, bait sequences from various species were collected for central proteins in the spliceosome, including pre-mRNA-processing factor 8 (PRP8), U1 small nuclear ribonucleoprotein C, U4/U6 small nuclear ribonucleoprotein Prp3, U4/U6.U5 small nuclear ribonucleoprotein 27 kDa protein, and U5 small nuclear ribonucleoprotein. Putative homologues in all animal species were detected based on Python scripts [57] and subjected to the construction of phylogenetic trees, as described previously [58].

Genome sequences were systematically screened for U12 spliceosome hints via Infernal (cmscan) v1.1.2 [59] with Rfam13 [60]. U4atac, U6atac, U11, and U12 were considered as indications for the presence of the minor U12 spliceosome in the respective species. Due to high computational costs, only a random subset of all animal sequences was analyzed.

### 2.5. Correlation between the GC Content of the Genome and the GC Content of the Splice Sites

The Pearson correlation coefficient between the GC content of the genome sequence of each species and the GC content of the respective splice site combination was calculated using the Python package scipy. Splice site combinations were weighted with the number of occurrences for assessment of the GC content. Finally, the correlation coefficient and the *p*-value were determined. For better visualization, a scatter plot was constructed with the Python package plotly [54].

### 2.6. Phylogeny of Non-Canonical Splice Sites

All *A. thaliana* transcripts with non-canonical splice sites were subjected to BLASTn v.2.8.1 searches against the transcript sequences of all other plant species previously studied [10]. The best hit per species was selected for an alignment against the respective genomic region with exonerate [61]. Next, splice site combinations were extracted and aligned. This alignment utilized MAFFT v7 [62] by representing different splice site combinations as amino acids. Finally, splice site combinations aligned with the non-canonical splice site combinations of *A. thaliana* were analyzed [45].

All transcripts of the fungus *Armillaria gallica* with non-canonical splice sites were searched as translated peptide sequences against all other fungal peptide sequence datasets via BLASTp [55]. Cases with more than 10 best hits with non-canonical splice site combinations in other species were subjected to the construction of phylogenetic trees for manual inspection. MAFFT v7 [62] and FastTree v2 [63] were used for the alignment and tree construction.

### 2.7. Usage of Non-Canonical Splice Sites

Genes were classified based on the presence/absence of non-canonical splice combinations into four groups: GT-AG, GC-AG, AT-AC, and minor non-canonical splice site genes. When having different non-canonical splice sites, genes were assigned into multiple groups. Next, the transcription of these genes was quantified using featureCounts v1.5.0-p3 [64] on the RNA-Seq read mapping generated with STAR v.2.5.1b. Multi-mapped reads were excluded from the analysis and expression values were calculated at gene level (-t gene). Binning of the genes was performed based on the fragments per kilobase transcript length per million assigned fragments (FPKMs). Despite various shortcomings [65], we considered FPKMs to be acceptable for this analysis. Outlier genes with extremely high values were excluded from this analysis and the visualization. Next, a cumulative sum of the relative bin sizes was calculated. The aim was to compare the transcriptional activity of genes with different splice site combinations, i.e., to test whether non-canonical splice site combinations are enriched in lowly transcribed genes.

Usage of splice sites was calculated per intron, as previously described [10]. The difference between both ends of an intron was calculated. The distribution of these differences per splice site type was analyzed.

Introns were grouped by their splice site combination. The average of both coverage values of the directly flanking exon positions was calculated as the estimate of the local expression around a splice site combination. Next, the sequencing coverage of a transcript was estimated by multiplying 200 bp (assuming 2 × 100 nt reads) with the number of read counts per gene and normalization to the transcript length. The difference between both values was calculated for each intron to assess its presence in the major isoform.

### 2.8. Genomic Read Mapping and Variant Calling

Genomic sequencing reads were retrieved from the SRA via fastq-dump, as described above. BWA MEM v.0.7 [66] was applied with the –M parameter for mapping of the reads, and GATK v.3.8 [67,68] was used for variant detection, as described previously [69]. The positions of variants were compared to the positions of splice sites using compare_variation_rates.py [10].

### 2.9. Data Availability

This work was based on publicly available datasets retrieved from the NCBI (Appendix A) and the SRA (Appendix A). Python scripts and a detailed technical description are available at GitHub [10,70]. Datasets with information about the coverage around splice sites in animals [46] and fungi [47] were made available as data publications at Bielefeld University Library.

## 3. Results

### 3.1. Analysis of Non-Canonical Splice Sites

In total, 64,756,412 (Appendix A) and 2,302,340 (Appendix A) splice site combinations in animals and fungi, respectively, were investigated based on annotated genome sequences (Appendix A). The average frequency of the canonical splice site combination GT-AG is 98.3% in animals and 98.7% in fungi, respectively. These values exceed the 97.9% previously reported for plants [10], thus the analyzed datasets indicate a generally higher frequency of non-canonical splice site combinations in plants.

Average percentages of the most important splice site combinations were summarized per kingdom and over all analyzed genomes (Table 1, Figure 1). The number of canonical and non-canonical splice site combinations per species was also summarized (Appendix A). A significantly (*p* ≈ 3.5 × 10^−14^, Mann-Whitney U-test) higher percentage of non-canonical splice sites was observed in animals in comparison to fungi. Further statistical comparisons of the splice site combination frequencies provided in Table 1 are summarized in Appendix A. Several species strongly exceeded the average values for major and minor non-canonical splice sites. The fungal species *Meyerozyma guilliermondi* shows approximately 6.67% major and 13.33% minor non-canonical splice sites. *Eurytemora affinis* (copepod) and *Oikopleura dioica* (tunicate) reveal approximately 10% minor non-canonical splice sites (Appendix A). The average frequency of sequence variants at splice sites is far below 1% (Appendix A). Although non-canonical splice sites are generally more likely to harbor sequence variants than canonical variants, these sequence variants can only account for a very small proportion of non-canonical splice site combinations.

Different properties of the genome sequences of all investigated species were analyzed to identify potential explanations for the observed differences in splice site frequencies (Appendix A). In fungi, the average number of introns per gene is 1.49 and the average GC content is 47.1% (±7.39; s.d.). In animals, each gene contains on average 6.95 introns and the average GC content is 39.4% (±3.87; s.d.). The average number of introns per gene in plants is 4.15 and the average GC content 36.3% (±8.84; s.d.).

A comparison of the genome-wide GC content to the GC content of all splice sites revealed a weak correlation in the analyzed fungal genomes (r ≈ 0.236, *p* ≈ 0.008). Species with a high genomic GC content tend to show a high GC content in the splice site combinations in the respective species. A similar correlation was found in plant (r ≈ 0.403, *p* ≈ 4.505 × 10^−6^) and animal species (r ≈ 0.434, *p* ≈ 7.866 × 10^−24^) (Appendix A). Additionally, the GC content in fungal genomes is substantially exceeding the average GC content of plant and animal genomes. Since genomic GC content and intronic GC content strongly correlate (animals: r ≈ 0.968, *p* ≈ 2.357×10^−292^; plants: r ≈ 0.974, *p* ≈ 8.987 × 10^−79;^ and fungi: r ≈ 0.950, *p* ≈ 2.800 × 10^−66^), the results obtained in the analysis above are representative for both methods of GC content calculation (Appendix A). Since splicing of U12 introns, which often shows the major non-canonical AT-AC splice site combination, requires the presence of the minor U12 spliceosome, we screened the genome sequences of all investigated species for components of this spliceosome. As differences in the genome sequence completeness and continuity, as well as sequence divergence from bait sequences, can impact the results, we only stated the presence of the U12 spliceosome in some species, while the absence in the remaining species could not be demonstrated. The comparison of annotated AT-AC splice site combinations between species with and without the minor U12 spliceosome revealed significantly higher numbers of this major non-canonical splice site combination in species with U12 spliceosomes (Mann-Whitney U-Test: *p* ≈ 3.8 × 10^−12^ (plants) and *p* ≈ 1.8 × 10^−15^ (animals)). Although many fungi are known to have a minor U12 spliceosome [19], we only detected corresponding RNA genes in one species (*Cutaneotrichosporon oleaginosum*) and thus refrained from any conclusions about the situation in fungi.

Based on the datasets analyzed here, the frequencies of the most frequent non-canonical splice site combinations differ between animals, fungi, and plants (Figure 1). In fungal genome sequences, the splice site combination CT-AC is substantially more frequent than the splice site combination AT-AC. Regarding the splice site combination GA-AG in animal genome sequences, two outliers are clearly visible: *E. affinis* and *O. dioica* show more GA-AG splice site combinations than GC-AG splice site combinations.

Despite overall similarities in the pattern of non-canonical splice site combinations between kingdoms, specific minor non-canonical splice site combinations were identified at a much higher frequency in some fungal and animal species. First, RNA-Seq data was harnessed to validate these unexpected splice site combinations. Next, the frequencies of selected splice site combinations across all species of the respective kingdom were calculated. The correlation between the size of the incorporated RNA-Seq datasets and the number of supported splice sites was examined as well (Appendix A). In animals, there is a correlation (r ≈ 0.417, p ≈ 0.022) between the number of supported splice sites and the total number of sequenced nucleotides in RNA-Seq data. For fungi, no correlation between the number of supported splice sites and size of the RNA-Seq datasets could be observed. It is important to note that the number of available RNA-Seq datasets from fungi was substantially lower. Further, analyses of introns with canonical and non-canonical splice site combinations, respectively, revealed that a higher number of introns is generally associated with a higher proportion of non-canonical splice sites (Appendix A).

### 3.2. High Diversity of Non-Canonical Splice Sites in Animals

Based on the analyzed datasets, substantial differences in the diversity of splice site combinations other than GT-AG and GC-AG in fungi (H’ ≈ 0.0277) and animals (H’ ≈ 0.0637) were observed (Kruskal-Wallis: *p* ≈ 0.00000). Besides the overall high proportion of minor non-canonical splice site combinations (Table 1), differences between species are high (Figure 1). The slightly higher interquartile range of splice site combination frequencies in animal species, and especially in plant species (Figure 1a,c), together with the relatively high frequency of “other” splice sites in animals and plants (Table 1), suggest more variation of splice sites in the kingdoms of animals and plants compared to the investigated fungal species. The high diversity of splice sites could be associated with the higher complexity of animal and plant genomes. In addition, the difference in prevalence between the major non-canonical splice site combination GC-AG and minor non-canonical splice site combinations is smaller in animals compared to fungi and plants (Figure 1).

GA-AG is a frequent non-canonical splice site combination in some animal species. Two species, namely *E. affinis* and *O. dioica*, showed a much higher abundance of GA-AG splice site combinations compared to the other investigated species (Figure 1a, Appendix A). RNA-Seq reads support 5795 (23% (average)) of all GA-AG splice site combinations of both species. GA-AG splice sites are supported in all analyzed species with a slightly lower frequency of 19%. In *E. affinis* and *O. dioica*, the number of the GA-AG splice site combination exceeds the number of the major non-canonical splice site combination GC-AG.

We quantified the proportion of GA-AG splice site combinations to 3.2% (5345) of all 166,392 supported splice site combinations in this species. The 5′ splice site GA is flanked by highly conserved upstream AG and a downstream A (Figure 2). Both species, *E. affinis* (Figure 2a,b) and *O. dioica* (Figure 2c,d), show striking similarities at several positions of 5′ and 3′ splice sites, in addition to the GA-AG dinucleotides.

### 3.3. CT-AC is a Frequent Splice Site Combination in Fungal Annotations

Although the general frequency pattern of fungal splice site combinations is similar to plants and animals, several fungal species displayed a high frequency of apparent minor non-canonical CT-AC splice site combinations. This co-occurs with a lower frequency of AT-AC splice site combinations. Our findings indicate that the minor non-canonical splice site combination CT-AC occurs with a significantly (Mann-Whitney U-Test; *p* ≈ 0.00035) higher frequency in the annotation of fungal genome sequences than the major non-canonical splice site combination AT-AC. In contrast, the frequency of AT-AC in animals (*p* ≈ 9.560 × 10^−10^) and plants (*p* ≈ 5.464 × 10^−24^) exceeds the CT-AC frequency significantly (Figure 3a). For the splice site combination CT-AC, a sequence logo, which shows the conservation of this splice site in four selected species, was designed (Figure 3b). The highest frequencies of the splice site combination CT-AC, supported by RNA-Seq reads, were observed in *Alternaria alternata, Aspergillus brasiliensis, Fomitopsis pinicola,* and *Zymoseptoria tritici* (approx. 0.08–0.09%). It is important to note that CT-AC resembles the canonical GT-AG splice site combination on the complementary strand. In addition to CT-AC, the flanking nucleotides also displayed complementary nucleotides of a canonical splice site combination (Figure 3b).

### 3.4. Intron Size Analysis

Assuming that non-canonical splice sites are not used or are used at a lower efficiency, it could be assumed that introns with non-canonical splice sites are more often retained than introns with canonical splice sites. A possible consequence of intron retention could be frameshifts unless the intron length is a multiple of three. Therefore, we investigated a total of 8,060,924, 737,783 and 2,785,484 transcripts across animals, fungi, and plants, respectively, with respect to their intron lengths. Introns with a length divisible by three (codon length) could be kept in the final transcript without causing a shift in the reading frame. In addition, these introns need to be free of any in-frame stop codons to permit intron retention. The proportion of introns with a length divisible by three and free of in-frame stop codons differs between introns with different splice site combinations (Table 2). While the values are higher for introns with non-canonical splice site combinations compared to canonical introns, there is no substantial difference between introns with major or minor non-canonical splice site combinations. Fungi are more likely to harbor introns with a length divisible by three and free of in-frame stop codons than animals or plants. This is likely due to generally shorter introns in fungal genes (Appendix A, [10]).

### 3.5. Conservation of Non-Canonical Splice Site Combinations across Species

In total, *A. thaliana* transcripts containing 1,073 GC-AG, 64 AT-AC, and 19 minor non-canonical splice sites were aligned to transcripts of all plant species. Homologous intron positions were checked for non-canonical splice sites. GC-AG splice site combinations were conserved in 9830 sequences, matched with other non-canonical splice site combinations in 121 cases, and aligned to GT-AG in 13,045 sequences. Given that the dominance of GT-AG splice sites was around 98%, the number observed here indicates a strong conservation of GC-AG splice site combinations. AT-AC splice site combinations were conserved in 967 other sequences, matched with other non-canonical splice site combinations in 93 cases, and aligned to GT-AG in 157 sequences. These numbers indicate a conservation of AT-AC splice site combinations, which exceeds the conservation of GC-AG splice site combinations substantially. Minor non-canonical splice sites were conserved in 48 other sequences, matched with other non-canonical splice site combinations in 64 cases, and were aligned to a canonical GT-AG splice site in 213 cases. This pattern suggests that most non-canonical splice site combinations are either (A) mutations of the canonical ones or (B) mutated towards GT-AG splice site combinations.

We also investigated non-canonical splice sites in transcripts of *Armillaria gallica*, as this species shows a high number of non-canonical splice sites in the annotation and the set of fungal genome sequences has a feasible size for this analysis. After identification of putative homologous sequences in other species, phylogenetic trees (Appendix A) of these sequences were inspected. Transcripts with non-canonical splice site combinations are clustered in clades that also harbor transcripts without non-canonical splice site combinations. We analyzed trees of all transcripts that have similar transcripts with non-canonical splice site combinations in at least 10 other species and observed on average five transcripts with a non-canonical splice site combination among the 10 closest relatives. This number exceeds the expectation based on the overall frequency of less than 3% non-canonical splice site combinations, thus indicating conservation of non-canonical splice sites.

### 3.6. Usage of Non-Canonical Splice Sites

To analyze a possible correlation of non-canonical splice sites with low transcriptional activity, we compared the transcript abundance of genes with non-canonical splice site combinations to genes with only canonical GT-AG splice site combinations (Figure 4a). Genes with at least one non-canonical splice site combination are generally less likely to be lowly expressed than genes with only canonical splice sites. While this trend holds true for all analyzed non-canonical splice site combination groups, GC-AG and AT-AC containing genes display especially low proportions of genes with low FPKMs.

To understand the amount of flexibility in respect to different terminal dinucleotides, we compared the usage of 5′ and 3′ splice sites over 4,141,196 introns in plants, 3,915,559 introns in animals, and 340,619 introns in fungi (Figure 4b). The plot shows that the 3′ splice site seems to be more flexible than the 5′ splice site, which was observed in all three kingdoms.

To evaluate the relevance of such alternative isoforms, we assessed the contribution of isoforms to the overall abundance of transcripts of a gene. Therefore, the usage of splice sites flanking an intron was compared to the average usage of splice sites. This reveals how often a certain intron is removed by splicing. Introns with low usage values might only be removed from minor transcript isoforms. While most introns display no or very small differences, GT-AG introns deviate from this trend. This indicates that non-canonical splice site combinations are frequently part of the dominant isoform. Again, these findings were similar for all of the investigated kingdoms.

## 4. Discussion

### 4.1. Analysis of Non-Canonical Splice Sites

In summary, the observed frequencies of canonical and major non-canonical splice site combinations are similar to the pattern previously reported for plants [10], but some essential differences and exceptions were found in animals and fungi based on the datasets analyzed here. As our strict filtering of splice site combination candidates based on RNA-Seq reads leads to an underestimation of the true number of non-canonical splice sites per species, all comparisons are based on the proportion rather than total numbers. Previous studies already revealed that non-canonical splice site combinations are not just the result of sequencing errors [9,10,21,33,71]. Here, we investigated the position of sequence variants in plants, fungi, and animals, with respect to splice sites. The average frequencies of canonical and non-canonical splice sites were calculated for each kingdom. The canonical splice site combination GT-AG occurs with a frequency of 98.3% in animals, 98.7% in fungi, and 97.9% in plants. These findings are in line with previous reports for selected species [21], but provide a higher resolution due to the genome-wide analysis. For example, the previously observed variation of the proportion of GC-AG splice site combinations multiplied by a factor of two [21] matches the center of the frequency distribution displayed in Figure 1. Proportions reported for GC-AG (<1%) and AT-AC (<0.01%) in model organisms [21] are in the same range as our findings. As previously speculated [10], a generally less accurate splicing system in plants could be an adaptation to changing environments through the generation of a larger transcript diversity. Since most plants are not able to change their geographic location, the tolerance for unfavorable conditions should be stronger than in animals.

The lower proportion of non-canonical splice site combinations in fungi compared to animals seems to contradict this hypothesis. However, the genome size and complexity need to be considered here. Based on the available assemblies, the average animal genome is significantly larger than the average fungal genome (Mann-Whitney U-Test; *p* = 5.64 × 10^−68^) (Appendix A). Although the average animal genome sequence (median = 998 Mbp) is longer than the plant average (median = 467 Mbp), plant genome sequences harbor more non-canonical splice site combinations (Appendix A, [10]).

Another property, the difference in the GC content between fungi (higher) and animals/plants (lower), could be associated with the much lower frequency of AT-AC splice site combinations in fungi (Figure 1). The AT-AC splice site combination has a very low GC content (25%). A generally higher GC content in a genome could result in higher GC content within splice site combinations due to the overall mutation rates in these species.

### 4.2. High Diversity of Non-Canonical Splice Sites in Animals

Kupfer et al. suggested that splicing may differ between fungi and vertebrates [29]. In a few animal species, especially in *E. affinis*, a high frequency of the non-canonical splice site combination GA-AG was detected. For *E. affinis*, the high frequency of the GA-AG splice site combinations was described previously when GA-AG was detected in 36 introns [72]. As the arthropod *E. affinis* [73] and the chordate *O. dioica* [74] belong to different phyla of the animal kingdom, the conservation of sequences flanking the 5′ and 3′ splice sites and the ability to splice GA-AG introns might be explained by (A) convergent evolution or (B) an ancestral trait, which was only preserved in a few species, including *E. affinis* and *O. dioica*.

Possible mechanisms, which could explain these GA-AG splice site combinations, are RNA editing or template switching by a reverse transcriptase. A high GC content of non-canonical splice site combinations, which is not valid for GA-AG splice sites, could facilitate the formation of secondary structures, ultimately leading to template switching [22]. However, RNA editing can lead to the formation of canonical splice sites on RNA level, even though a non-canonical splice site is present on DNA level [40].

Efficient splicing of the splice site combination GA-AG was detected in human fibroblast growth factor receptor genes [75]. Further, it was suggested that this splicing event is, among other sequence properties, dependent on a canonical splice site six nucleotides upstream [75], which does not exist in the genome sequence of the species investigated here (Figure 2). An analysis of all five potential U1 snRNAs in *E. affinis* did reveal one single nucleotide polymorphism in the binding site of the 5′ splice site from C to U at position eight in one of these U1 snRNAs. This could result in the binding of pre-mRNAs originating from A/GATAAGT instead of AG/GTAAGT (Figure 5) [72,76]. Although this would imply an elegant way for the splicing of GA-AG splice site combinations, the same variation was also detected in a putative human U1 snRNA, which was identified via BLAST against the human reference genome sequence (NC_000001.11). This human U1 snRNA is annotated as a pseudogene (GeneID:106480160). Therefore, we speculate that this gene might be active and functional in *E. affinis*, while it has lost its activity in other lineages. However, another mechanism or additional factors might be involved in splicing of introns containing the GA-AG splice site combination. Although a modified copy of spliceosomal components is a likely explanation for the observed GA-AG splice site combination, no higher amplification rate of spliceosome parts was observed in *E. affinis* and *O. dioica* compared to other animal species.

### 4.3. CT-AC is a Frequent Splice Site Combination in Fungal Annotations

Non-canonical splice sites in fungi were, so far, only described in studies that focused on a single or a few species. An analysis in the fungus-like microorganism *Phytophthora sojae* [77,78] revealed 3.4% non-canonical splice site combinations GC-AG and CT-AC [79]. Our findings in this systematic study show that CT-AC appears as a major non-canonical splice site combination in fungi, while AT-AC occurs with a much lower frequency than in animals and plants (Figure 2). However, the presence of antisense transcripts, which would be spliced at a canonical GT-AG splice site combination (reverse complement of CT-AC), is a very likely explanation. At least, stranded RNA-Seq datasets are required to investigate this hypothesis in fungi by differentiating between transcripts of both strands. Frequently encountered artifacts caused by reverse transcription [22] might be avoided in the future through direct RNA sequencing [80]. Due to the very limited availability of suitable datasets for fungal species we had to leave this question for future studies.

### 4.4. Conservation of Non-Canonical Splice Site Combinations Across Species

Non-canonical splice site combinations frequently aligned to splice site combinations of the same type in plant and fungal species. However, the power of this analysis is currently limited by the quality of the alignment. Although splice site combinations should be aligned properly in most cases, small differences in the number could be caused by ambiguous situations. Due to this apparently complex evolutionary pattern, we do not know if these clades originated from (I) a canonical splice site combination that accumulated mutations multiple times or (II) a non-canonical splice site combination that was turned into a canonical one multiple times. To explain each non-canonical splice site combination by one of these possibilities, a tool for automatic inspection of the observed phylogenetic pattern would be required. It is likely that both hypotheses stated above account for a fraction of splice site combinations. A previous analysis [10] revealed that minor non-canonical splice site combinations are present in genes with various levels of conservation. No clear enrichment in a certain group of genes was reported [10]. As comprehensive functional annotations are not yet available for most of the studied species, we refrained from enrichment analysis. However, the presence of non-canonical splice sites in specific gene groups should be addressed in the future based on suitable annotations.

### 4.5. Usage of Non-Canonical Splice Sites

Non-canonical splice site combinations were described to have regulatory roles by slowing down the splicing process [81]. Previous reports also indicated that non-canonical splice site combinations might appear in pseudogenes [9,10]. We speculate that a stronger transcriptional activity of genes with non-canonical splice sites compensates for lower turnover rates in the splicing process. The regulation of these genes might be shifted from the transcriptional to the post-transcriptional level. This trend is similar for animals and plants (Appendix A). In fungi, genes with minor non-canonical splice sites display relatively high proportions of genes with low FPKMs. Moreover, a higher number of non-canonical splice sites per gene is associated with a lower expression. This leads to the suggestion that non-canonical splice sites occur more often within pseudogenes. Further, our observations regarding usage of the 5′ and 3′ splice sites align well with previous findings of a higher flexibility at the 3′ splice site compared to the 5′ splice site [82]. A mutated 5′ splice site represses the removal of the upstream intron [10,83,84]. Further, for plants and animals, the difference between the usage of the 5′ splice site and the 3′ splice site is notably higher for introns with the splice site combination GC-AG. Although bona fide non-canonical splice site combinations are present in many plant transcripts [10], additional transcript isoforms might exist. Previously reported NAGNAGs could be an additional explanation for this observation as alternative 3′ splice sites could be used without causing a frameshift [82].

One important limitation of this investigation is the sparse availability of stranded RNA-Seq datasets and direct RNA sequencing datasets. Therefore, it is not possible to rule out the involvement of antisense transcripts, which have been observed before [85]. In three cases, these antisense transcripts could be spliced at a canonical or major non-canonical splice site combination, while appearing as a minor non-canonical splice site on the investigated strand. Previous reports of antisense transcripts [86] support this explanation for some of our observations. However, most of the observed minor non-canonical splice site combinations cannot be explained by antisense transcripts as these would also harbor a minor non-canonical splice site combination.

## 5. Conclusions

Our investigation of non-canonical splice sites in animal, fungal, and plant genome sequences revealed kingdom-specific differences, which were supported by the analyzed RNA-Seq data. Animal species with a high proportion of annotated GA-AG splice site combinations were examined. Further, properties of introns and splice sites were analyzed. Consistently across all kingdoms, the 3′ splice site appeared to be more flexible than the 5′ splice site. Across fungal genome sequences, the splice site combination CT-AC is more frequent than the splice site combination AT-AC. This turns CT-AC into an apparent major non-canonical splice site combination in fungal species, while AT-AC should be considered a minor non-canonical splice site there. However, this high frequency of CT-AC in fungal genome sequence annotations might be due to the presence of antisense transcripts, which are properly spliced at a canonical GT-AG splice site combination (the reverse complement of CT-AC). Overall, our findings demonstrate the importance of considering non-canonical splice sites despite their low relative frequency in comparison to the canonical splice site combination GT-AG. RNA-Seq data supported the existence and usage of numerous non-canonical splice site combinations. By neglecting non-canonical splice sites in gene prediction processes, bona fide genes might be excluded or at least structurally altered.

## Figures and Tables

**Figure 1 cells-09-00458-f001:**
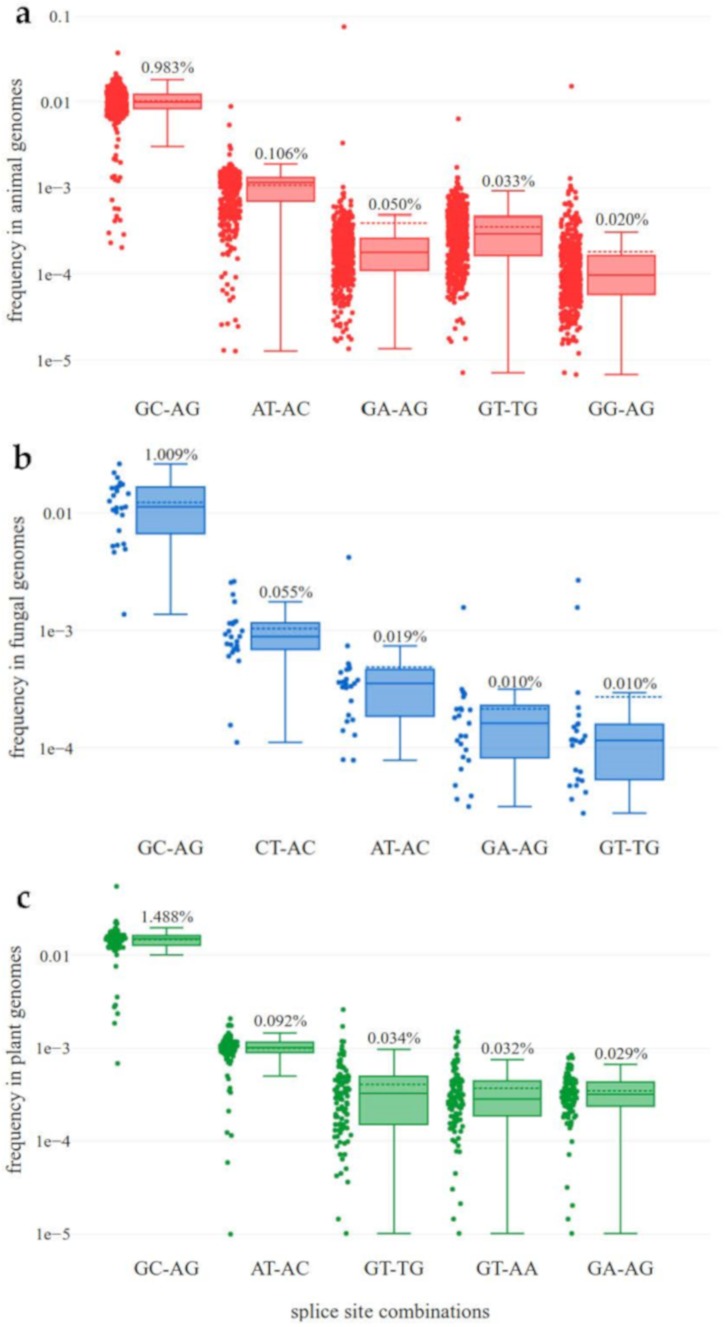
Frequencies of non-canonical splice site combinations in animals, fungi, and plants. The frequency of non-canonical splice site combinations across the 489 animal (red, **a**), 130 fungal (blue, **b**), and 121 plant (green, **c**) genome sequences is shown. Normalization of the absolute number of each splice site combination was performed per species based on the total number of annotated splice site combinations in representative transcripts. The frequency of the respective splice site combination of each species is shown on the left-hand side and the percentage of the respective splice site combination is shown on top of each box plot. The dashed line represents the mean frequency of the respective splice site combination over all investigated species. The box plots are ordered (from left to right) according to the mean frequency.

**Figure 2 cells-09-00458-f002:**
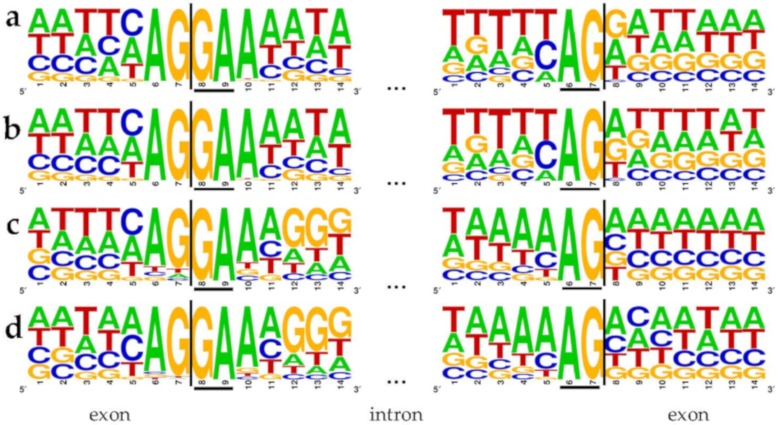
Flanking positions of GA-AG splice site combinations in *Eurytemora affinis* (**a**,**b**) and *Oikopleura dioica* (**c**,**d**). All splice site combinations (**a**,**c**) as well as all 5795 with RNA-Seq data supported splice site combinations (**b**,**d**) of these two species were investigated. Seven exonic and seven intronic positions are displayed at the 5′ and 3′ splice sites. Underlined bases represent the terminal dinucleotides of the intron, i.e., the 5′ and 3′ splice site.

**Figure 3 cells-09-00458-f003:**
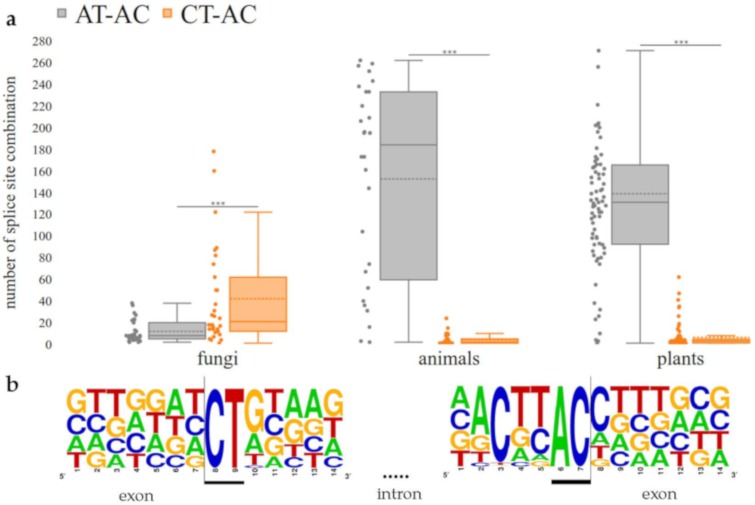
CT-AC frequency exceeds AT-AC frequency in the annotation of fungal genome sequences. (**a**) Number of the minor non-canonical CT-AC splice site combination in comparison to the major non-canonical splice site combination AT-AC in each kingdom (Mann-Whitney U-Test; fungi: *p* ≈ 0.00035, animals: *p* ≈ 9.560 × 10^−10^, plants: *p* ≈ 5.464 × 10^−24^). The dashed line represents the mean frequency of the respective splice site combination over all investigated species. (**b**) Sequence logo for the splice site combination CT-AC in four selected fungal species (*Alternaria alternata, Aspergillus brasiliensis, Fomitopsis pinicola* and *Zymoseptoria tritici*). In total, 67 supported splice sites with this combination were used to generate the sequence logo.

**Figure 4 cells-09-00458-f004:**
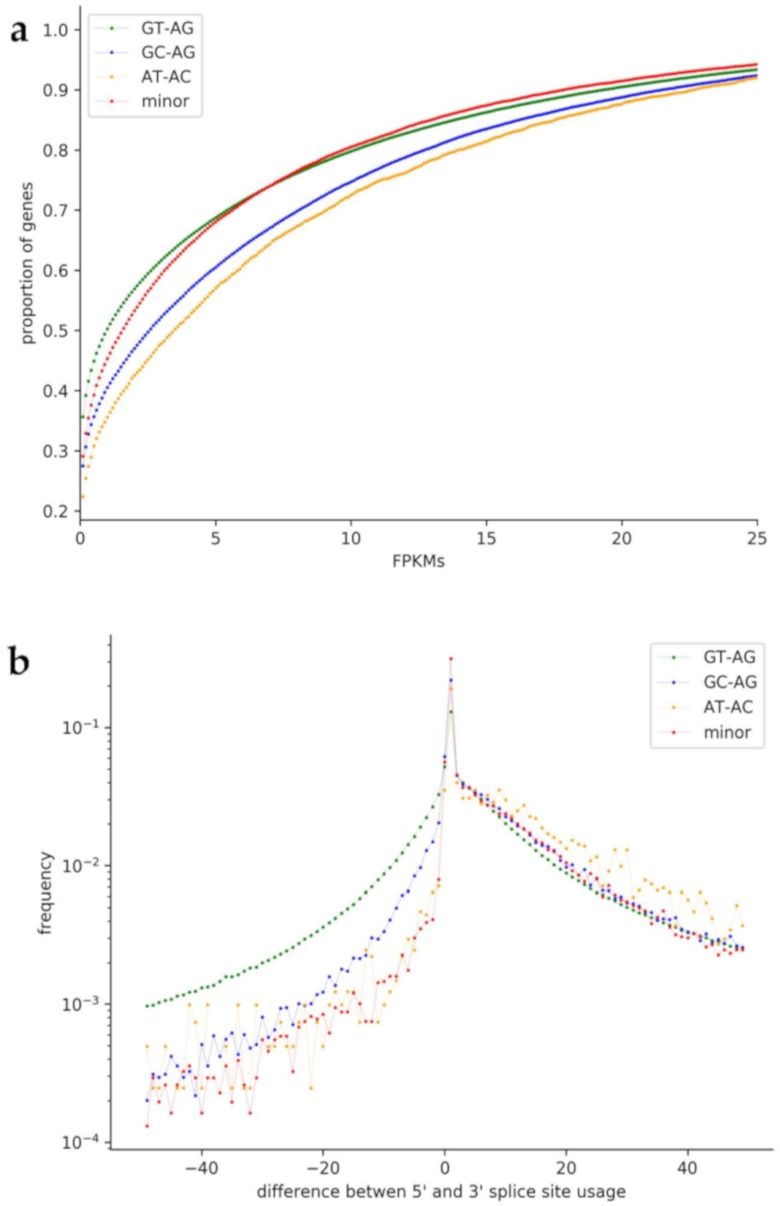
Usage of non-canonical splice site combinations in plant species. (**a**) Comparison of the transcript abundance (FPKMs) of genes with non-canonical splice site combinations to genes with only canonical GT-AG splice site combinations. GC-AG and AT-AC containing genes display especially low proportions of genes with low FPKMs. (**b**) Comparison of the usage of 5′ and 3′ splice sites. On the x-axis, the difference between the 5′ splice site usage and the usage of the 3′ splice site is shown. A fast drop of values when going to the negative side of the x-axis indicates that the 3′ splice site is probably more flexible than the 5′ splice site.

**Figure 5 cells-09-00458-f005:**
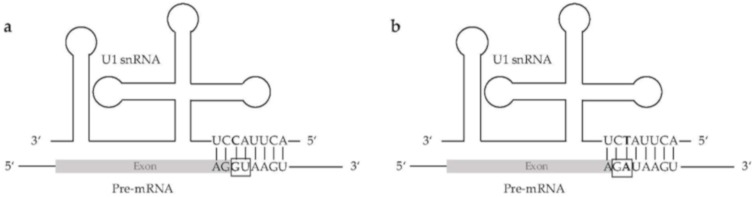
Hypothetical binding of the U1 snRNA to the pre-mRNA. (**a**) Binding sequence of the canonical U1 snRNA to the canonical 5′ splice site GU (GT on DNA). (**b**) Hypothetical binding sequence of the non-canonical U1 snRNA (C > T) to the non-canonical 5′ splice site GA.

**Table 1 cells-09-00458-t001:** Splice site combination frequencies in animals, fungi, and plants. Only the most frequent combinations are displayed here, and all minor non-canonical splice site combinations are summarized as one group (“Others”). A full list of all splice site combinations is available (Appendix A).

	GT-AG	GC-AG	AT-AC	Others
Animals	98.334%	0.983%	0.106%	0.577%
Fungi	98.715%	1.009%	0.019%	0.257%
Plants	97.886%	1.488%	0.092%	0.534%
All	98.265%	1.074%	0.101%	0.560%

**Table 2 cells-09-00458-t002:** Proportion of introns with a length divisible by three and without an in-frame stop codon. The results of intron length analysis for selected splice site combinations for animals, fungi, and plants are shown.

	Splice Site Combination	Introns Divisible by Three and without In-Frame Stop Codon	Number of Introns Divisible by Three and without In-Frame Stop Codon
Animals	GT-AG	1.4%	*n* = 922561
	AT-AC	2.8%	*n* = 1945
	GC-AG	3.0%	*n* = 19166
	Others	4.1%	*n* = 15554
Fungi	GT-AG	9.2%	*n* = 208522
	AT-AC	10.7%	*n* = 46
	GC-AG	13.3%	*n* = 3096
	Others	13.1%	*n* = 785
Plants	GT-AG	2.7%	*n* = 382538
	AT-AC	3.6%	*n* = 10356
	GC-AG	4.8%	*n* = 478
	Others	5.8%	*n* = 4481

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
