# Peer review of "Animal, Fungi, and Plant Genome Sequences Harbor Different Non-Canonical Splice Sites"

_cells, 2020, doi:10.3390/cells9020458_

Round 1

Reviewer 1 Report

Frey and Pucker discuss alternative (non-canonical) splice sites in several organisms comparing organism dependency. The manuscript is well developed and organized but with general pour attention to the style.

Major comments

Please revise all the manuscript using the same style for titles and text. E.g. Title 2.9 has a different font size from 3.1; text from line 253 has a different font size from the previous text. Is there a statistically significant difference among the same splicing sites between animal fungi and plants or are similar? (not statistically different). Please include this information in table 1. Authors say: “The most frequent non-canonical splice site combinations show differences between animals, fungi, and plants” Please calculate the statistical significance of displayed differences in figure 1 and include results in the figure. Organize the manuscript including figures after there are cited in the text. For example, figure 1 is showed before its mention in the text. Authors say: “Two species, namely E. affinis and O. dioica, showed a much higher abundance of GA-AG splice site combinations compared to the other investigated species (Figure 1A).” This reviewer does not understand the correspondence to figure 1A. How can I retrieve the information for the two species? The authors compare the expression of genes with different splice sites. I am interested in the function of genes with the alternative usage of splice sites. Are there functionally correlated among organisms and within the animal, plants, and fungi?

Reviewer 2 Report

This manuscript describes a bioinformatics meta-analysis of large RNA-seq datasets to derive the frequency of non-canonical splice sites in three eukaryotic kingdoms such as plants, animals and fungi. Major trends of non-canonical 5’ and 3’ splice sites as well as particular cases specific to few species are derived, to conclude that non-canonical splice sites are slightly more prevalent in plants, and that the relative frequencies of non-canonical splice sites are different across kingdoms. Overall this study adds to previous reports of non-canonical splice sites which are well cited with few exceptions mentioned below, and whose citations should be added. Also this reviewer has concerns about measuring non-canonical splice sites as many of these were reported to be artifacts or used with a very low frequency and perhaps non-functional. Details of these criticisms are provided below, and I would just recommend this article for publication once a strong and convincing rebuttal and revised version is re-submitted.

My main concern is about the reliability of the data (not the authors’ fault), with the limitation to make comparisons across eukaryotic kingdoms with samples of diverse origins and characteristics. For instance, authors show that non-canonical splice sites tend to be found in highly-expressed genes, while this suggests that these non-canonical sites might just be rare by-products of splicing with no functional significance. The authors mention that fungi RNA-seq datasets have less coverage, etc, which might limit the interpretations. Also some non-canonical splice sites may map to pseudogenes, as hinted later. Hence, the conclusion that plants have a higher incidence of non-canonical splice sites might be caused by the heterogeneous underlying RNA-seq data.

Authors mention that non-canonical splice sites in non-mammals were estimated from mammals (page 3). A study reported non-canonical (GC-AG, U12-type) splice sites in worms and fruitflies, in addition to human, mouse and A thaliana (Sheth et al NAR 2006 PMID=16914448). The data in the current paper should be compared to this relevant study as well.

For AT-AC non-canonical splice sites, authors should mention that a small subset of these are U2-dependent and named AT-AC type II introns, as initially reported in Wu et al RNA 1997 (PMID= 9174094), and more recently in Kubota et al Hum Mutat 2011 (PMID= 21412952).

The CT-AC non-canonical splice sites in fungal species are clearly artefacts from mapping exons and introns to the antisense strand when RNA-seq does not have strand specificity. First, the reverse complement of CT-AC is GT-AG. Second, the logos in Figure 3 bottom right show a consensus of ACTTAC/CT which is the direct reverse complement of the AG/GTAAGT, which is the exact consensus for canonical GT 5’ splice sites. To this reviewer, barring experimental confirmation, this is not a coincidence!

To me, the analysis of intron length divisibility by 3 in Table 2, showing no obvious differences, is not very meaningful. This divisibility relates to the frame-shifting that can be caused by intron retention, while authors ignored that intron retention can cause premature termination because of in-frame stop codons in the intron, which occur more frequently by random chance the longer the introns are. A normalization to intron length should be added in this analysis.

The diversity or flexibility of splice-site usage is higher for 3’ splice sites as expected, so authors here should mention NAGNAGs, which are tandem 3’ splice site AGs used alternatively, see Yan et al FEBs Let 2015 (PMID= 26028313).

The frequent GA-AG non-canonical splice sites in some fungi is an interesting finding that could be attributed to the corresponding mutant U1 snRNA in these species. However, authors mention (page 15) that humans also have such mutation (with no citation or further info!), which is very surprising. What is this mutant U1, is it one of its many pseudogenes? Clarify.

Minor amendments: On page 1, splicing can occur during or after transcription. I also suggest to relabel ‘donor’ and ‘acceptor’ with their correct and unambiguous nomenclature, which is 5’ splice site and 3’ splice site. Clarify this sentence on page 3: “Over 40 % of plant genes with exactly 40 exons are affected [10]”. Page 13: “After identification of putative homologous sequences in other species, phylogenetic trees of these sequences were inspected”, where are these trees??

Round 2

Reviewer 1 Report

Authors replayed to all my comments satisfactory. I have no other comments and the manuscript can be considered for the publication.

Reviewer 2 Report

All ok now.